# Effects of Cerium Oxide on Kidney and Liver Tissue Damage in an Experimental Myocardial Ischemia-Reperfusion Model of Distant Organ Damage

**DOI:** 10.3390/medicina60122044

**Published:** 2024-12-11

**Authors:** Işın Güneş, Ali Doğan Dursun, Çağrı Özdemir, Ayşegül Küçük, Şaban Cem Sezen, Mustafa Arslan, Abdullah Özer

**Affiliations:** 1Department of Anesthesiology and Reanimation, Erciyes University Faculty of Medicine, Kayseri 38039, Turkey; iekinci@erciyes.edu.tr; 2Department of Physiology, Atılım University Faculty of Medicine, Ankara 06560, Turkey; alidogandursun@gmail.com; 3Vocational School of Health Services, Atilim University, Cankaya, Ankara 06805, Turkey; 4Home Care Services, Medicana International Ankara Hospital, Cankaya, Ankara 06520, Turkey; 5Department of Anesthesiology and Reanimation, Gazi University Faculty of Medicine, Ankara 06560, Turkey; mdcagriozdemir@gmail.com; 6Department of Physiology, Kutahya Health Sciences University Faculty of Medicine, Kutahya 43100, Turkey; aysegul.kucuk@ksbu.edu.tr; 7Department of Histology and Embryology, Kırıkkale University Faculty of Medicine, Kırıkkale 71000, Turkey; sezenscem@gmail.com; 8Application and Research Centre for Life Sciences, Gazi University, Ankara 06560, Turkey; 9Centre for Laboratory Animal Breeding and Experimental Research (GÜDAM), Gazi University, Ankara 06560, Turkey; 10Department Cardiovascular Surgery, Gazi University Faculty of Medicine, Ankara 06560, Turkey; dr-abdozer@hotmail.com

**Keywords:** myocardial ischemia reperfusion, cerium oxide, kidney, liver, TOS, TAS

## Abstract

*Background and Objectives:* Ischemia-reperfusion (I/R) injury is a process in which impaired perfusion is restored by restoring blood flow and tissue recirculation. Nanomedicine uses cutting-edge technologies that emerge from interdisciplinary influences. In the literature, there are very few in vivo and in vitro studies on how cerium oxide (CeO_2_) affects systemic anti-inflammatory response and inflammation. Therefore, in our study, we aimed to investigate whether CeO_2_ administration has a protective effect against myocardial I/R injury in the liver and kidneys. *Materials and Methods:* Twenty-four rats were randomly divided into four groups after obtaining approval from an ethics committee. A control (group C), cerium oxide (group CO), IR (group IR), and Cerium oxide-IR (CO-IR group) groups were formed. Intraperitoneal CeO_2_ was administered at a dose of 0.5 mg/kg 30 min before left thoracotomy and left main coronary (LAD) ligation, and myocardial muscle ischemia was induced for 30 min. After LAD ligation was removed, reperfusion was performed for 120 min. All rats were euthanized using ketamine, and blood was collected. Liver and kidney tissue samples were evaluated histopathologically. Serum AST (aspartate aminotransferase), ALT (alanine aminotransaminase), GGT (gamma-glutamyl transferase), glucose, TOS (*Total Oxidant Status*), and TAS (*Total Antioxidant Status*) levels were also measured. *Results:* Necrotic cell and mononuclear cell infiltration in the liver parenchyma of rats in the IR group was observed to be significantly increased compared to the other groups. Hepatocyte degeneration was greater in the IR group compared to groups C and CO. Vascular vacuolization and hypertrophy, tubular degeneration, and necrosis were increased in the kidney tissue of the IR group compared to the other groups. Tubular dilatation was significantly higher in the IR group than in the C and CO groups. TOS was significantly higher in all groups than in the IR group (*p* < 0.0001, *p* < 0.0001, and *p* = 0.006, respectively). However, TAS level was lower in the IR group than in the other groups (*p* = 0.002, *p* = 0.020, and *p* = 0.031, respectively). Renal and liver histopathological findings decreased significantly in the CO-IR group compared to the IR group. A decrease in the TOS level and an increase in the TAS level were found compared to the IR group. The AST, ALT, GGT, and Glucose levels are shown. *Conclusions:* CeO_2_ administered before ischemia-reperfusion reduced oxidative stress and ameliorated IR-induced damage in distant organs. We suggest that CeO_2_ exerts protective effects in the myocardial IR model.

## 1. Introduction 

According to the World Health Organization, approximately 2.3 million people die annually from ischemic heart disease [1,2]. Myocardial ischemia results from inadequate coronary perfusion. Ischemia, defined as insufficient oxygen delivery to tissues, causes tissue damage as a result of the consumption of energy stores and the inability to excrete metabolites. Anaerobic metabolism is also observed. The treatment involves achieving reperfusion as soon as possible.

Paradoxically, reperfusion causes more serious tissue damage than ischemia [3,4,5]. It causes distant organ damage by activating oxidative stress such as inflammation and ischemia. Increased vasodilation and apoptosis exacerbate acute tissue damage. Calcium accumulation, cell necrosis, edema, and free reactive oxygen species (ROS) occur. The ability to deliver oxygen to the ischemic tissue with reperfusion triggers oxidative reactions. Products with significant toxic potential, such as ROS, are formed. As a secondary effect of reperfusion, these products pass into the systemic circulation. Under normal conditions, antioxidant mechanisms maintain the ROS balance in organisms by removing them from the environment. In cases where antioxidant defense capacity is exceeded, ROS increase, and this often means serious tissue damage [5]. The damage that occurs is not limited to the organ where ischemia occurs but also occurs in distant organs such as the kidney and liver.

Exogenous antioxidant therapy may be necessary in cases where the oxidant–antioxidant balance shifts towards oxidative stress. Clearing excess ROS and supporting endogenous antioxidant defenses are strategies used to minimize cellular damage. Recently, nanoparticles have attracted attention because of their antioxidant properties [6]. Cerium oxide nanoparticles (CeO_2_) are widely used in ultraviolet absorbance, oxygen sensing, and automotive catalytic converters. Previous studies have highlighted the antioxidant activity of CeO_2_ nanoparticles. It has been shown that it can provide protection by reducing oxidative stress in the primary tissue and distant organs in different ischemia reperfusion (I/R) models. This antioxidant property has been attributed to CeO_2_′s ability to mimic endogenous antioxidant cellular defense [7,8,9].

In this study conducted on rats, the effect of intraperitoneal cerium oxide administered before left main coronary artery ischemia on kidney and liver damage due to myocardial ischemia reperfusion injury was investigated.

## 2. Materials and Methods 

### 2.1. Experimental Protocol

The study was conducted in the GUDAM Laboratory of Gazi University with the consent of the Experimental Animal Ethics Committee of Gazi University (G.Ü.E.T-21-022). Experimental procedures were performed in accordance with the standards of the Guide for the Care and Use of Laboratory Animals. Twenty-four male Wistar Albino rats, weighing 250–275 g, were used. They were kept at 20–21 °C in 12 h daylight/12 h darkness. They had free access to food until 2 h before anesthesia.

The animals were randomly separated into four groups, each containing six rats. The control group (Group C), cerium oxide group (Group CO), ischemia-reperfusion (Group IR), and CO and CO+(I/R) (Group CO-IR) groups were formed. Rats underwent left thoracotomy, and 0.5 mg/kg CeO_2_ (Co aqueous nanoparticle dispersion, 100 mL; Sigma-Aldrich; Merck KGaA, Darmstadt, Germany) was administered intraperitoneally (i.p.) 30 min before ligating the LAD [10]. A small plastic snare was threaded through the ligature and was placed in contact with the heart. The artery was then occluded by applying tension to the ligature (30 min), and reperfusion was achieved by releasing the tension (120 min). For anesthesia, 50 mg/kg (i.p.) ketamine (500 mg/10 ml Ketalar^®^; Parke-Davis, Detroit, MI, USA; Pfizer, Inc., New York, NY, USA) and 10 mg/kg (i.p.) xylazine (Alfazyne^®^ 2%; Alfasan International B.V., Woerden, The Netherlands) were preferred. The cardiac effects of ketamine were optimized by administering the same amount of ketamine to all rats.

After anesthesia, the rats were placed in a supine position, and the trachea was cannulated for ventilation. A left thoracotomy was performed, and the fourth and fifth ribs were cut to the left of the sternum. Positive pressure artificial respiration was immediately started with room air at a volume of 1.5 mL/100 g body weight and a rate of 60 beats per minute. Then, 500 IU/kg sodium heparin was administered via the tail vein. After the pericardium was cut, the heart was removed. An 8/0 silk suture was quickly placed under the left main coronary artery. During reperfusion, the thorax was covered with a sterile moist pad.

If a reaction to surgical stimuli was observed, 20 mg/kg ketamine was administered to deepen anesthesia. Following the completion of the IR model, 100 mg/kg ketamine was administered to all rats for sacrifice, and 5–10 mL of intracardiac blood was collected. The kidneys and liver were removed for biochemical and histopathological analyses. Tissues to be used for histopathological examination were placed in 10% formalin, while tissues to be used for biochemical examination were frozen in liquid nitrogen and stored at −80 °C.

A graphical representation is presented in Figure 1.

### 2.2. Histopathological Analysis

The tissues were fixed with 10% formalin for 48 h at room temperature and embedded in paraffin. Then, 4 µm thick sections were taken from the paraffin blocks and stained with hematoxylin and eosin. The sections were scanned end-to-end.

### 2.3. Liver

Hydropic degeneration, pyknotic nuclei, necrosis and mononuclear cellular infiltration, and sinusoidal dilatation were investigated. The semi-quantitative evaluation technique used for histological testing by Abdel-Wahhab et al. [11] was used to interpret the structural changes in hepatic tissues; a negative point (0) indicated no structural changes, one positive point (1, +) indicated mild changes, two positive points (2, ++) indicated moderate structural changes, and three positive points (3, +++) indicated severe structural changes [12].

### 2.4. Kidney

Glomerular vacuolization (GV), vascular vacuolization and hypertrophy (VVH), tubular dilatation (TD), cell degeneration and necrosis (TCDN), hyaline casts (THC), cell shedding (TCS), Bowman space dilatation (BSD), and lymphocyte infiltration (LI) were evaluated (29). Kidney damage was scored as follows: 0, no change; +1, minimal change; +2, moderate; +3, severe [13].

### 2.5. Homogenization of Tissues/Measurements of TOS/TAS

TOS (Total Oxidant Status) and TAS (Total Antioxidant Status) homogenization and measurements were studied with the method specified in previous publications [13,14,15]. 

### 2.6. Serum Alanine Aminotransaminase (ALT), Aspartate Aminotransferase (AST), Gamma-Glutamyl Transferase (GGT), Glucose Levels

The serum levels of aminotransferases (AST and ALT), GGT, and glucose were measured using routine laboratory tests.

### 2.7. Statistical Analysis

The results obtained were processed by variance analysis using the Statistical Package for the Social Sciences (SPSS, Chicago, IL, USA) 22.0 program. Shapiro–Wilk and Q–Q plot tests were used to evaluate data distribution. Biochemical and histopathological parameters were tested using the Kruskal–Wallis test. A significant Kruskal–Wallis test was used to determine which group was different from the others, followed by the Mann–Whitney U test with the Bonferroni adjustment. Data were expressed as standard error (mean ± SE) and mean ± standard deviation (SD). Differences were considered statistically significant at *p* < 0.05.

## 3. Results

### 3.1. Histopathological Results

On histopathological examination, the damage ranged from none (control and CO groups) to mild (CO-IR group) and severe (I/R group). No findings suggestive of tissue damage were observed in the livers of the rats in the control group. The liver tissues of rats in the CO group were similar to group C, and no damage was observed. When the liver tissues of the rats in the I/R group were examined, as a result of the myocardial I/R model, hepatocyte degeneration, necrotic cells, and mononuclear cell infiltration in the parenchyma were found to be significantly higher than the other groups. In the CO-IR group, necrotic cell and mononuclear cell infiltration was significantly reduced compared to I/R, but no significant improvement was seen in hepatocyte degeneration. When the CO-IR, C, and CO groups were compared, no significant differences were observed (Table 1, Figure 2, Figure 3, Figure 4, Figure 5 and Figure 6).

Histopathological examination of rat kidney tissues showed that the damage ranged from none (control and CO groups) to mild (CO-IR group) to severe (I/R group). The kidney tissues of the rats in the control group showed normal histological features. The kidney tissues of rats in the CO group were similar to group C, and no CeO_2_-induced damage was observed. After the myocardial I/R model was established, tubular dilatation, vascular vacuolization and hypertrophy, and tubular cell degeneration and necrosis were significantly different between the groups, all of which were increased in the IR group. Vascular vacuolization and hypertrophy and tubular cell necrosis and degeneration in the CO-IR group administered CeO_2_ were significantly reduced compared with those in the IR group, and the results were similar to those in the C and CO groups. Tubular dilatation was higher in the IR and CO-IR groups than in the other two groups. Although dilation decreased in rats in the CO-IR group compared to the IR group, this difference was not statistically significant (Table 2, Figure 7, Figure 8, Figure 9, Figure 10 and Figure 11).

### 3.2. Biochemical Results 

The TOS levels were significantly higher in all groups compared to the IR group (*p* < 0.001, *p* < 0.0001, and *p* = 0.006, respectively). TOS levels in the CO-IR group were significantly lower than those in the I/R group and were similar to those in the CO group. However, TAS levels were higher in all groups than in the IR group (*p* = 0.002, *p* = 0.020, and *p* = 0.031, respectively). In the CO-IR group, the TAS level increased significantly according to I/R group and was similar to that in the CO and C groups. When oxidative status was evaluated according to the TOS and TAS parameters, the model-induced oxidative balance in the rats in the IR group was directed towards oxidative stress. Regarding TAS, CeO_2_ strengthened the antioxidant defense and brought it to levels close to those of the C and CO groups. However, the fact that TOS was higher in the CO-IR group than in the control group may indicate that oxidative balance was not established. The OSI results supported the TOS and TAS data. The highest OSI value was in the IR group. The OSI value decreased in the cerium oxide applied groups but remained higher in the CO-IR group than in the C group (Table 3, Figure 12).

Serum AST levels increased in the I/R group. The change in AST levels in the CO-IR group, where CeO_2_ was applied, did not differ significantly from that in the IR group. However, the AST levels in the CO-IR group were not higher than those in the C and CO groups (Table 3, Figure 13).

ALT and GGT levels were high in the IR group. These values were similar to those of the other three groups. This can be interpreted as a positive effect of CeO_2_ treatment on ALT and GGT levels (Table 3, Figure 13).

The glucose levels were higher in the IR group than in the control group. The serum glucose levels of the CO-IR and CO groups treated with CeO_2_ were lower than those of the IR group. However, the serum glucose levels in the CeO_2_ groups were higher than those in group C (Table 3, Figure 13).

## 4. Discussion

Cardiovascular diseases are one of the leading causes of death worldwide. The chief among them is coronary artery disease. There is a decrease in peripheral blood flow during myocardial ischemia, and a subsequent increase in reperfusion [16,17]. This can damage organs with high blood flow, such as the kidneys and the liver. Healthy kidneys and the liver consume relatively large amounts of oxygen, which corresponds to approximately 10% of the body’s total oxygen consumption. Therefore, abnormal ROS formation occurs prominently in these tissues. The ROS produced are metabolized by adaptive scavenging mechanisms. However, excessive ROS production may lead to acute or progressive tissue damage [18]. Based on the literature, supporting the antioxidant defense system may be an important treatment for reducing damage. This study aimed to contribute to the limited data in the literature on reducing distant organ damage by evaluating the effects of CeO_2_ 30 min before myocardial I/R on kidney and liver damage. Therefore, we examined both histopathological and oxidative antioxidative parameters in IR muscle tissue to investigate the effects of CeO_2_ on tissue damage. In the present study, we showed that intraperitoneal CeO_2_ application in a myocardial I/R model significantly reduced oxidative damage in the kidney and liver tissues and improved histopathological findings.

Recent developments in nanoparticles have increased their application areas, making them an increasingly popular scientific subject. CeO_2_ has attracted the attention of researchers owing to its unique oxidation state [19]. Cerium oxide contains positively charged holes on its surface owing to the gaps in the d” and f” orbitals. The electropositive charge of the surface indicates the antioxidant nature of the oxide nanoparticles. Cerium oxide nanoparticles effectively bind free radical species, hydroxyl radicals, hydrogen peroxide, superoxide anion radicals, and singlet oxygen and reduce oxidative stress. The antioxidant nature of cerium oxide makes it effective against diseases caused by oxidative stress [20,21,22,23]. There are different hypotheses regarding the antioxidant properties of cerium oxides. One of them is the reversibility between cerium atoms in the +3 and +4 oxidation states on their surfaces. This redox dynamic property allows cerium oxides to mimic the function of superoxide dismutase, catalyzing the conversion of superoxide anion to oxygen and hydrogen peroxide, the latter being subsequently reduced to water, thus reducing oxidative damage to cells [24]. In addition, some regions on the surface of cerium oxides serve as platforms for capturing and transferring electrons, further enhancing their antioxidant effects [25]. Cerium oxide can promote the dissociation and nuclear translocation of the Nrf2 transcription factor, an important regulator of the cellular defense mechanism that activates the expression of various antioxidant response genes and can strengthen the cell’s own defense mechanisms [26,27]. Activation of the Nrf2 pathway by cerium oxide increases the expression of endogenous antioxidant enzymes such as superoxide dismutase, catalase, and glutathione peroxidase, which play a direct role in scavenging ROS. Cerium oxides also increase the biosynthesis of the antioxidant molecule glutathione, which is the primary non-enzymatic antioxidant in cells and is crucial for neutralizing ROS and protecting cells from oxidative damage [28]. In addition, cerium oxides can regulate various signaling pathways involved in ROS scavenging, including modulation of intracellular calcium signaling and the MAPK signaling pathway, which are closely related to the cellular antioxidant response [29].

We included this unique nanoparticle in our study with the hypothesis that it would reduce kidney and liver damage due to myocardial ischemia-reperfusion, due to its high antioxidant potential. Özdemirkan et al. [30] investigated the protective effect of CeO_2_ on lung injury before lower extremity IR injury and observed that histopathological inflammation and biochemically oxidative stress decreased. The results obtained from Tuncay et al. [31] in a similar model supported this. Based on these results, we applied 0.5 mg/kg i.p. CeO_2_ 30 min before ischemia. We preferred the intraperitoneal route due to its ease of application and rapid absorption [32].

Free radicals are highly reactive and react with all cell components in the kidney. These reactions can cause tissue damage via the oxidative reaction chains. IR causes ROS formation in organs. Ischemia reduces the activity of antioxidant enzymes against ROS, and reperfusion further disrupts the oxidant/antioxidant balance [33], causing intracellular calcium overload, adenosine triphosphate depletion, myocardial apoptosis, and endothelial dysfunction [34,35]. Oxidative stress is a state of oxidant-antioxidant imbalance caused by oxidants that exceed antioxidant capacity. TOS and TAS are markers of oxidative stress [36,37]. TOS is a global indicator of all ROS types and TAS is a global indicator of antioxidant defense. TOS and TAS may provide a more accurate interpretation in terms of evaluating changes in the oxidant-antioxidant balance. Studies have shown that TOS and TAS measurements provide more valuable information than individual measurements of the parameters [38,39].

Metabolism, detoxification, and maintenance of redox balance depend on a healthy liver [40]. To carry out these processes, hepatocytes contain between 1000 and 2000 mitochondria and occupy approximately 20% of the cell volume [41]. Mitochondria are vulnerable to oxidative damage due to significant ROS production. In the event of oxidative stress, calcium influx into cells can trigger apoptotic and necrotic death [42]. These responses increase mitochondrial permeability and facilitate the release of pro-apoptotic factors and the activation of calcium-dependent endonucleases, proteases, and lipases, contributing to hepatocyte death [43]. In addition, oxidative stress can impair bile flow and affect the secretory function of hepatocytes, leading to cholestasis [44]. Studies have shown that oxidative stress increases hepatocyte degeneration, sinusoidal dilatation, pyknotic nuclei, necrotic cells, and parenchymal mononuclear cell infiltration in the liver [12,45,46] These histopathological damage indicators did not increase equally in each study and, in some models, the increase was not statistically significant. To avoid being overlooked owing to this heterogeneity, all damage markers were examined in our study. When our results were examined, it was observed that these pathological indicators, hepatocyte degeneration, necrotic cells, and mononuclear cell infiltration in the parenchyma were significantly higher in rats in the I/R group. High serum TOS and TAS levels suggested that this damage was caused by oxidative stress. In fact, there was no increase in any of these damage indicators in the control group, where oxidative parameters were normal. As an indication that CeO_2_ strengthens the weakened antioxidant defense, an increase in TAS levels and a decrease in TOS levels were observed in rats administered CeO_2_. In the CO-IR group, necrotic cell and mononuclear cell infiltration was significantly reduced compared to I/R, but no significant improvement in hepatocyte degeneration was observed. To better understand parenchymal and biochemical damage in the liver and to support the results, alanine transaminase (ALT), aspartate aminotransferase (AST), and gamma-glutamyl transferase (GGT) levels were examined. Similar to previous results, serum AST, ALT, and GGT levels increased in rats in the I/R group. Although CeO_2_ treatment brought ALT and GGT levels close to those of the control group, a similar result was not observed for AST levels. Cerium oxide has been shown to reduce liver damage in similar ischemia/reperfusion models. The results obtained in our study support those of the previous studies.

Mitochondrial dysfunction in the kidneys is shown to be a key step in the early stages of ischemic kidney injury [47]. Sureshbabu found that increased ROS levels increase mitochondrial apoptosis and lead to cell damage [48]. Mitochondrial dysfunction results in increased ROS levels, which amplify the effects of hypoxia on the renal medulla [49]. The resulting damage has been assessed in several studies using glomerular vacuolization, tubular dilatation, vascular vacuolization and hypertrophy, tubular cell degeneration and necrosis, Bowman’s space dilatation, tubular hyaline casts, lymphocyte infiltration, and tubular cell shedding. Once acute kidney injury has occurred, supportive care becomes important [13,50]. Free radicals are highly reactive and participate in many oxidative reactions that cause tissue damage. Ischemia reduces the activity of cellular defense enzymes against ROS, and reperfusion further disrupts the oxidant/antioxidant balance [33], whereas intracellular calcium overload, adenosine triphosphate depletion, myocardial apoptosis, and endothelial dysfunction are observed in diseases [35]. In addition, oxidative products such as ROS, reactive nitrogen species, hydrochloric acid, MDA, and lipid peroxides also generate TOS [51]. Oxidative stress is the state of oxidant–antioxidant imbalance caused by oxidants exceeding antioxidant capacity. TOS and TAS are indicators of oxidative stress [36,37]. They can provide a more accurate interpretation in terms of evaluating changes in the oxidant–antioxidant balance. In this study, TAS and ROS-induced oxidative stress were used as markers of total antioxidant protection against ROS attack. TOS was used as a stress marker. It has been shown that TOS and TAS analysis provide more valuable information than individual measurements of other oxidative stress parameters [38,39].

Renal hypoperfusion is the most common cause of acute kidney injury [52]. Pre-renal kidney injury can cause disease damage, manifesting as ischemic acute tubular necrosis. Arm conditions that show proximal tubular and medullary thickness, which are metabolically active, are susceptible to hypoperfusion owing to high oxygen connections [53]. It has been reported in the literature that mitochondrial dysfunction, as in intelligence, is an important step in the early stage of ischemic kidney damage [47]. When conditions causing ischemia are eliminated, reperfusion therapy is further increased. Hydroxyl radicals such as peroxynitrite and hyperchlorous acid are produced as ROS. In addition, antioxidant enzymes such as superoxide dismutase, catalase, and glutathione reductase are also depleted. It is found in the renal tissue under ischemic conditions [54]. Sureshbabu et al. In a previous study, the authors found that increased ROS levels increased mitochondrial apoptosis and led to cell damage [48]. Mitochondrial dysfunction causes microvascular dysfunction, and an increase in hypoxia in the renal medulla causes an increase in ROS [49]. The resulting damage was evaluated based on glomerular vacuolization, tubular dilatation, vascular vacuolization and hypertrophy, tubular cell degeneration and necrosis, Bowman space-giving, tubular hyaline cylinders, lymphocyte infiltration, and tubular cell proliferation. Similar results have been obtained in previously published ischemia-reperfusion models [55,56]. One must also consider the limited therapeutic treatment for support, care, and prevention or treatment of acute kidney injury and the scope of preventive treatments [13,50]. In this study, CeO_2_ significantly reduced TOS production, increased the amount of TAS, and reduced I/R-induced damage. These results indicated that CeO_2_ has a beneficial effect on myocardial I/R-induced damage to distant organs.

The hypoglycemic effect of cerium oxide has been reported in previous studies. The hypoglycemic potential of cerium oxide was investigated by inhibiting the catalytic activity of enzymes involved in the digestion of disaccharides and polysaccharides to reduce glucose release in the body. Cerium oxide nanoparticles act as a competitive inhibitor in the enzymatic reaction and assume the role of antienzyme. This reduces the enzyme–substrate reaction and prevents enzymatic degradation [57]. In the experimental study on enzymes, cerium oxide created strong inhibition on alpha-amylase, alpha-glucosidase, and sucrase. These nanoparticles have a high affinity for glucose molecules, similar to metformin. It was discovered that it is useful in regulating glucose metabolism and that the underlying molecular mechanism for the treatment of hyperglycemia is found [58]. A similar effect was seen in our study. Serum glucose levels increased in the IR groups, but glucose levels decreased after cerium oxide application.

There were some limitations to our study. Hemodynamic parameters were not measured in our study. The effects of hemodynamic changes on oxidative stress cannot be excluded. The data obtained at the end of the experiment were descriptive. However, specific effect pathways were not examined in this study. The experiment included only 30 min ischemia and 2 h reperfusion period. The effects of CeO_2_ on oxidative stress may change during long I/R periods. A sample size/power analysis could not be performed due to the restriction imposed by the animal research committee on the number of animals allowed. Consequently, the number of rats in each group was determined based on the committee’s authorization.

The serum glucose levels were also assessed in this study. As expected, after myocardial I/R, the blood glucose levels increased. In the CO-IR group, the glucose level decreased significantly compared with that in the IR group. However, interestingly, the glucose level in the CO-IR group was higher than that in the control group. It was unusually higher in the CO group than that in the control group. When looking at the literature, it has been shown that CeO_2_ improves blood glucose levels. The fact that the rats in the CO-IR group had lower glucose levels than those in the IR group explains this, but the increase in the CO group is remarkable.

## 5. Conclusions

In our study, distant damage was demonstrated histopathologically in the kidney and liver after myocardial I/R injury. Similar results have been reported for previously published I/R models. Future studies that analyze the antioxidant activity of CeO_2_ in different models will provide more in-depth information, especially regarding its effect on reducing I/R-related tissue damage. Our data suggest that prophylactic treatment with CeO_2_ nanoparticles may be a new therapeutic strategy for the treatment of ischemia, especially in ischemic heart disease.

## Figures and Tables

**Figure 1 medicina-60-02044-f001:**
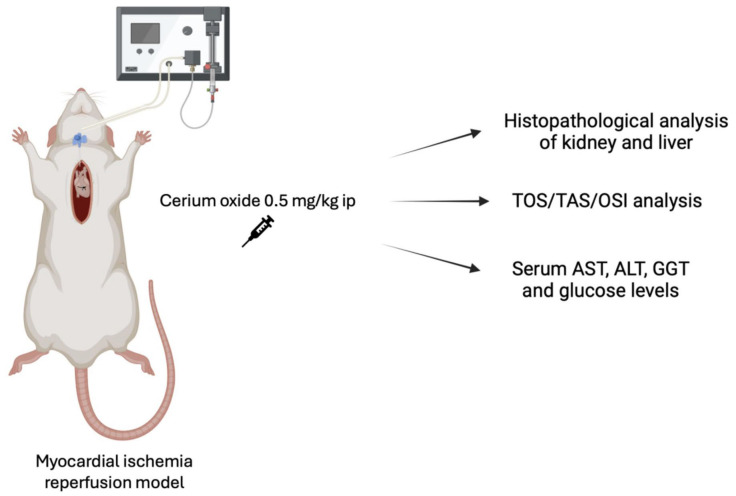
A graphical representation including the experimental procedure.

**Figure 2 medicina-60-02044-f002:**
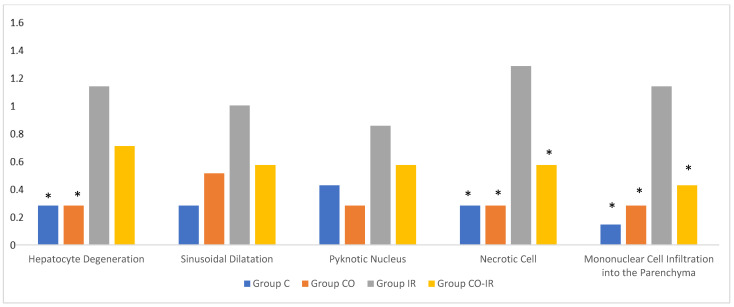
Rat Liver Tissue Histopathological Findings. * *p* < 0.5: Compared with Group IR; Group C: Control group; Group CO: Cerium oxide group; Group IR: Ischemia-reperfusion group; Group CO-IR: Cerium oxide-Ischemia-reperfusion group.

**Figure 3 medicina-60-02044-f003:**
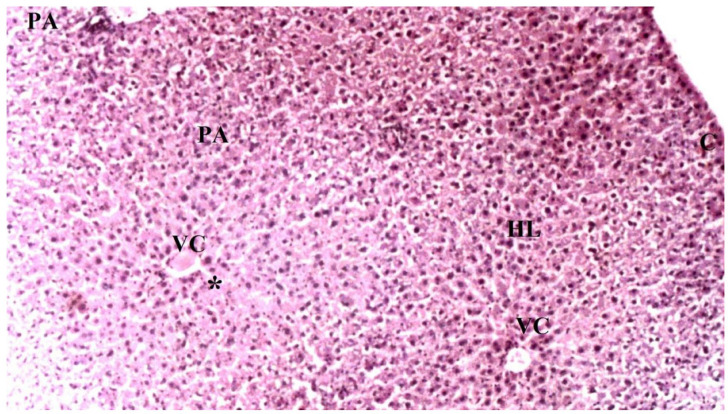
Representative light microscopy of liver tissue from the control group (PA: portal space, VC: vena centralis, *: sinusoids, HL: hepatic lobule), (H&EX40).

**Figure 4 medicina-60-02044-f004:**
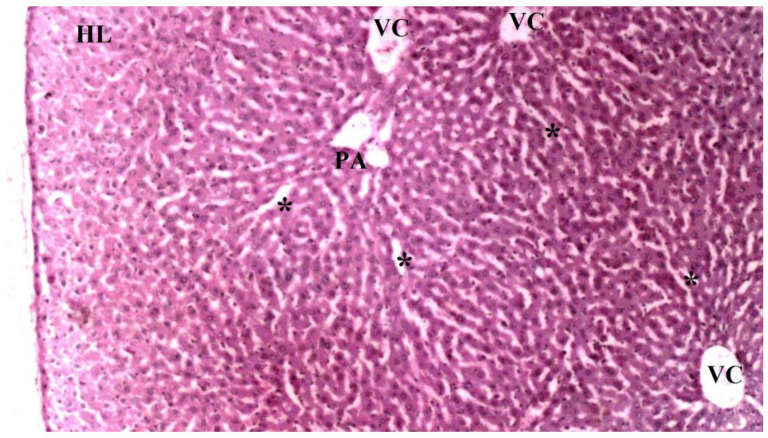
Representative light microscopy of liver tissue from the cerium oxide group (PA: portal space, VC: vena centralis, *: sinusoids, HL: hepatic lobule), (H&EX40).

**Figure 5 medicina-60-02044-f005:**
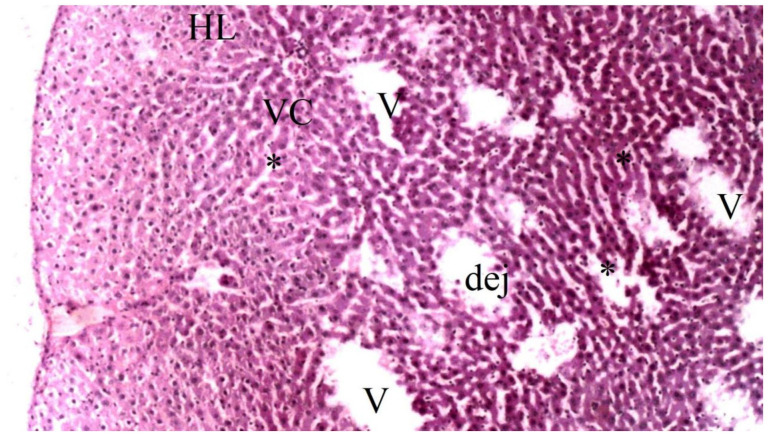
Representative light microscopy of liver tissue from the ischemia-reperfusion group (PA: portal space, VC: vena centralis, *: sinusoid dilatation, HL: hepatic lobule, dej: hepatocytes and vena centralis degeneration, V: portal vein), (H&EX40).

**Figure 6 medicina-60-02044-f006:**
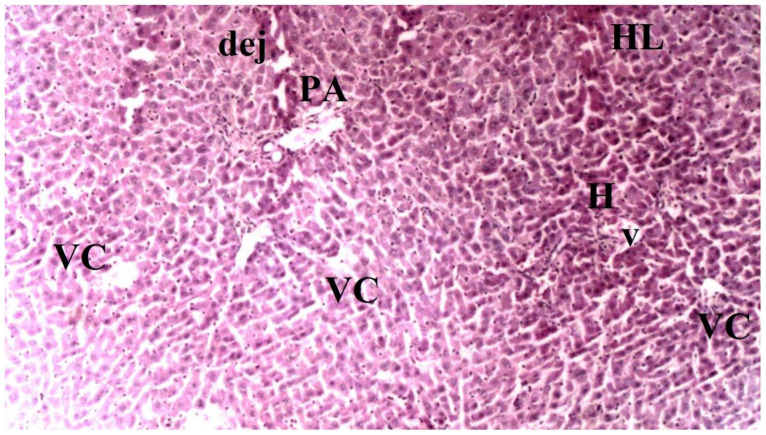
Representative light microscopy of liver tissue from cerium oxide-ischemia-reperfusion group (PA: portal space, VC: vena centralis, H: hepatocyte, HL: hepatic lobule, dej: hepatocytes and vena centralis degeneration, V: portal vein), (H&EX40).

**Figure 7 medicina-60-02044-f007:**
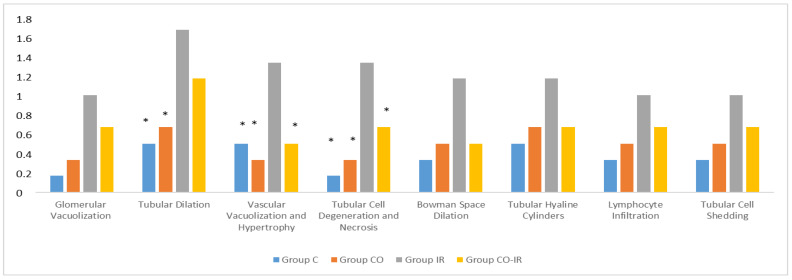
Rat Kidney Tissue Histopathological Findings. * *p* < 0.5: Compared with Group IR; Group C: Control group; Group CO: Cerium oxide group; Group IR: Ischemia-reperfusion group; Group CO-IR: Cerium oxide-Ischemia-reperfusion group.

**Figure 8 medicina-60-02044-f008:**
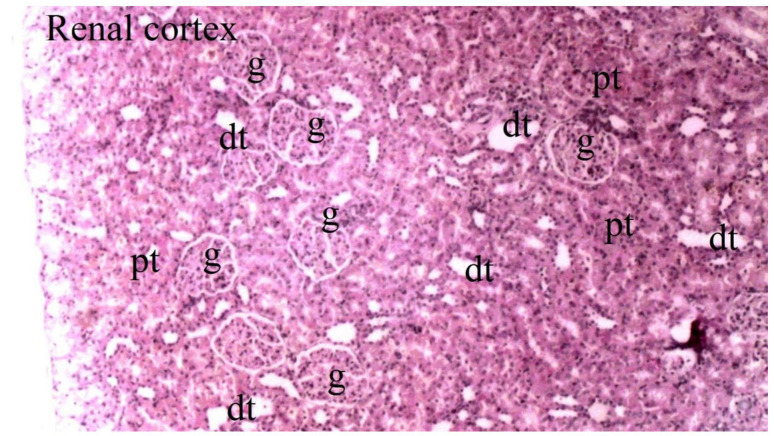
Representative light microscopy of kidney tissue from the control group (pt: proximal tubule, dt: distal tubule, g: glomerulus) (H&EX40).

**Figure 9 medicina-60-02044-f009:**
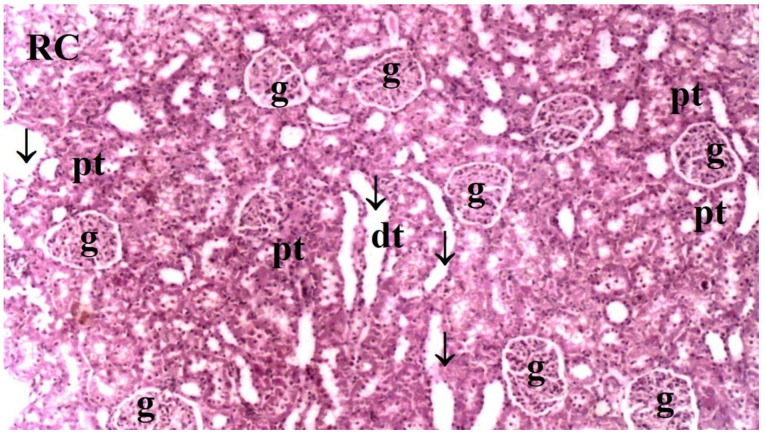
Representative light microscopy of kidney tissue from the cerium axide group (RC: renal cortex, pt: proximal tubule, dt: distal tubule, g: glomerulus, ↓: dilate tubule) (H&EX40).

**Figure 10 medicina-60-02044-f010:**
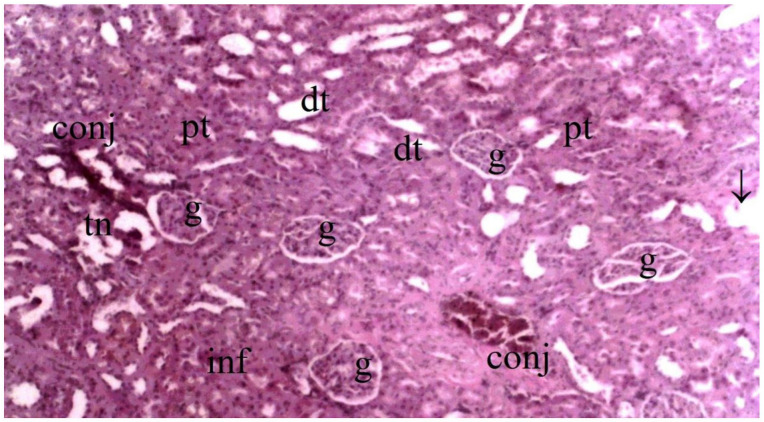
Representative light microscopy of kidney tissue from ischemia/reperfusion group (pt: proximal tubule, dt: distal tubule, g: glomerulus, m: macula densa, v: vacuolization, ↓: dilate tubule, conj: congestion, inf: inflammation, tn: tubular cell degeneration and necrosis) (H&EX40).

**Figure 11 medicina-60-02044-f011:**
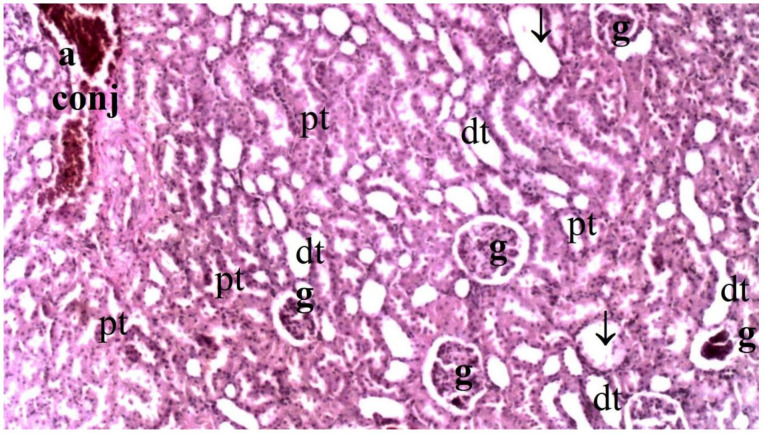
Representative light microscopy of kidney tissue from the cerium oxide ischemia-reperfusion group (pt: proximal tubule, dt: distal tubule, g: glomerulus, ↓: dilate tubule, conj: congestion, a: artery) (H&EX40).

**Figure 12 medicina-60-02044-f012:**
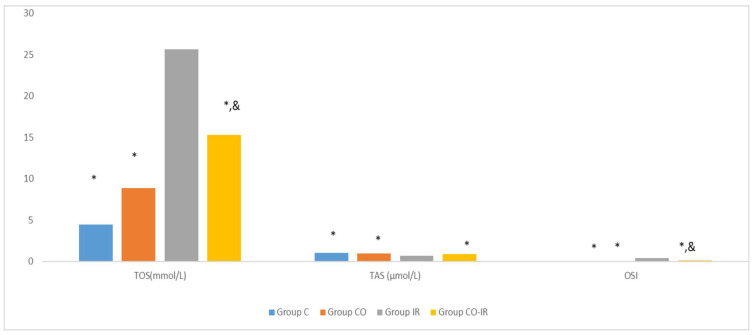
Rat Serum Total Oxidant Status, Total Antioxidant Status and Oxidative Stress Index. *: *p* < 0.05: Compared to Group IR; &: *p* < 0.05: Compared to Group C; Group C: Control group; Group CO: Cerium oxide group; Group IR: Ischemia-reperfusion group; Group CO-IR: Cerium oxide-Ischemia-reperfusion group.

**Figure 13 medicina-60-02044-f013:**
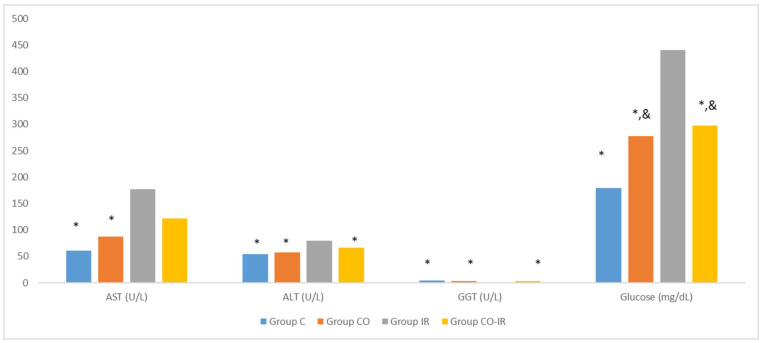
Rat Serum Alanine Aminotransaminase (ALT), Aspartate Aminotransferase (AST), gamma-glutamyl transferase (GGT), Glucose. *: *p* < 0.05: Compared to Group IR; &: *p* < 0.05: Compared to Group C; Group C: Control group; Group CO: Cerium oxide group; Group IR: Ischemia-reperfusion group; Group CO-IR: Cerium oxide-Ischemia-reperfusion group.

**Table 1 medicina-60-02044-t001:** Rat Liver Histopathological Findings [Mean ± SE].

	Group C (*N* = 6)	Group CO (*N* = 6)	Group IR (*N* = 6)	Group CO-IR (*N* = 6)	*p* **
**Hepatocyte Degeneration**	0.33 ± 0.21 *	0.33 ± 0.21 *	1.33 ± 0.33	0.83 ± 0.17	0.022
**Sinusoidal Dilatation**	0.33 ± 0.21	0.60 ± 0.25	1.17 ± 0.17	0.67 ± 0.21	0.060
**Pyknotic Nucleus**	0.50 ± 0.22	0.33 ± 0.21	1.00 ± 0.26	0.67 ± 0.21	0.226
**Necrotic Cell**	0.33 ± 0.21 *	0.33 ± 0.21 *	1.50 ± 0.34	0.67 ± 0.21 *	0.008
**Mononuclear Cell Infiltration into the Parenchyma**	0.17 ± 0.17 *	0.33 ± 0.21 *	1.33 ± 0.21	0.50 ± 0.34 *	0.013

* *p* < 0.05: Compared With Group IR; *p* **: Significance Level With Kruskal–Wallis Test *p* < 0.05; Group C: Control group; Group CO: Cerium oxide group; Group IR: Ischemia-reperfusion group; Group CO-IR: Cerium oxide-Ischemia-reperfusion group.

**Table 2 medicina-60-02044-t002:** Rat Kidney Histopathological Findings [Mean ± SE].

	Group C (*N* = 6)	Group CO (*N* = 6)	Group IR (*N* = 6)	Group CO-IR (*N* = 6)	*p* **
**Glomerular Vacuolization**	0.17 ± 0.17	0.33 ± 0.21	1.00 ± 0.36	0.67 ± 0.21	0.121
**Tubular Dilation**	0.50 ± 0.22 *	0.67 ± 0.21 *	1.67 ± 0.33	1.17 ± 0.17	0.012
**Vascular Vacuolization and Hypertrophy**	0.50 ± 0.22 *	0.33 ± 0.21 *	1.33 ± 0.21	0.50 ± 0.22 *	0.017
**Tubular Cell Degeneration and Necrosis**	0.17 ± 0.17 *	0.33 ± 0.21 *	1.33 ± 0.21	0.67 ± 0.21 *	0.003
**Bowman Space Dilation**	0.33 ± 0.21	0.50 ± 0.22	1.17 ± 0.31	0.50 ± 0.34	0.182
**Tubular Hyaline Cylinders**	0.50 ± 0.22	0.67 ± 0.21	1.17 ± 0.31	0.67 ± 0.21	0.264
**Lymphocyte Infiltration**	0.33 ± 0.21	0.50 ± 0.22	1.00 ± 0.00	0.67 ± 0.21	0.105
**Tubular Cell Shedding**	0.33 ± 0.21	0.50 ± 0.22	1.00 ± 0.26	0.67 ± 0.21	0.226

* *p* < 0.05: Compared With Group IR; *p* **: Significance Level With Kruskal Wallis Test *p* < 0.05; Group C: Control group; Group CO: Cerium oxide group; Group IR: Ischemia-reperfusion group; Group CO-IR: Cerium oxide-Ischemia-reperfusion group.

**Table 3 medicina-60-02044-t003:** Rat Serum Alanine Aminotransaminase (ALT), Aspartate Aminotransferase (AST), gamma-glutamyl transferase (GGT), Glucose, Total Oxidant Status, Total Antioxidant Status and Oxidative Stress Index [Mean ± SD].

	Group C (*N* = 6)	Group CO (*N* = 6)	Group IR (*N* = 6)	Group CO-IR (*N* = 6)	*p* **
**AST (U/L)**	60.56 ± 3.02 *	87.17 ± 10.85 *	177.33 ± 39.81	121.50 ± 15.84	0.008
**ALT (U/L)**	54.40 ± 4.20 *	57.83 ± 2.87 *	80.17 ± 4.57	66.50 ± 5.30 *	0.002
**GGT (U/L)**	4.36 ± 0.52 *	3.43 ± 0.44 *	1.69 ± 0.36	3.02 ± 0.28 *	0.002
**Glucose (mg/dL)**	179.33 ± 27.61 *	277.50 ± 22.25 *,&	439.83 ± 22.82	297.83 ± 23.40 *,&	<0.0001
**TOS (mmol/L)**	4.50 ± 0.77 *	8.88 ± 1.62 *	25.68 ± 3.86	15.28 ± 2.11 *,&	<0.0001
**TAS (µmol/L)**	1.04 ± 0.09 *	0.94 ± 0.04 *	0.69 ± 0.02	0.92 ± 0.11 *	0.014
**OSI**	0.04 ± 0.01 *	0.09 ± 0.05 *	0.38 ± 0.15	0.16 ± 0.03 *,&	<0.001

*p* **: Significance Level with Kruskal–Wallis Test *p* < 0.05; *: *p* < 0.05: Compared to Group IR; &: *p* < 0.05: Compared to Group C; Group C: Control group; Group CO: Cerium oxide group; Group IR: Ischemia-reperfusion group; Group CO-IR: Cerium oxide-Ischemia-reperfusion group.

## Data Availability

Data from the current study are available from the corresponding author upon request.

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
