# Peer review of "Effects of Cerium Oxide on Kidney and Liver Tissue Damage in an Experimental Myocardial Ischemia-Reperfusion Model of Distant Organ Damage"

_medicina, 2024, doi:10.3390/medicina60122044_

Round 1
Reviewer 1 Report
Comments and Suggestions for Authors
In this study, the authors tested a novel nanotechnology-based approach for treating myocardial I/R injury in the liver and kidneys using CeO2 nanoparticles (CeO2-NPs). These nanoparticles possess enzyme-like antioxidant properties that mitigate organ and tissue damage caused by oxygen free radicals. CeO2-NPs demonstrated protective effects in both in vitro and in vivo disease models by addressing multiple facets of I/R injury, including mitochondrial protection, antioxidant activity, reduced superoxide production, improved sensorimotor function, and reduced hepatic I/R damage.
Comments:
- All figures are missing in the attached manuscript.
- Please include a graphical representation of the methods used for CeO2 administration, left thoracotomy, left main coronary (LAD) ligation, and induction of myocardial ischemia.
- Present all data in bar graphs or suitable graphical formats, alongside the tables.
- Line 163 states that TAS levels were higher in the IR group than in the IR group, which is incorrect. TAS is actually higher in all other groups compared to the IR group. Specifically, TAS levels are higher in Groups C, CO, and CO-IR than in Group IR, while TOS is lower in Groups C, CO, and CO-IR than in Group IR. This does not imply that CeO2-NP improves antioxidant status, so it would be more appropriate to represent the TOS/TAS ratio along with the TOS and TAS data.
- Clarify which NOX or antioxidant mechanisms are enhanced by CeO2-NP treatment.
- CeO2-NP treatment decreases glucose levels in Group CO-IR compared to Group IR. Are these fasting glucose levels, and does CeO2-NP treatment improve insulin sensitivity?
- Although the study does not include any hematological parameters, is there evidence of cytotoxicity associated with nanoparticle treatment?
- Is there any cell-specific targeting of this nanoparticle to the liver or kidney?
Author Response
- Figures have been uploaded to the system.
- A graphical representation including the experimental procedure has been uploaded to the system.
- Bar Graphs containing all data have been presented.
- Errors noticed by you in the biochemical result section have been corrected in the text. The corrections are stated in the text. TAS levels were at similar levels except for the IR group. The fact that TAS levels decreased with IR were similar in other groups was interpreted by us as a strengthening of antioxidant defense. The absence of the TAS decrease that should have occurred after IR in the CO-IR group cannot be explained in any other way. A similar situation is valid for TOS levels. However, in the cerium oxide applied The fact that TOS levels in the CO-IR group were higher than the control group indicates that oxidative stress was not reduced sufficiently. OSI data is added to the table. Statistical interpretation of OSI data has been added.
- Hypotheses about the mechanisms by which cerium oxide strengthens antioxidant defense are explained in the discussion section with references 24-29.
- In our study, rats were fed ad libitum. No fasting was applied. The hypoglycemic effect of cerium oxide has been reported in previous studies. The hypoglycemic potential of cerium oxide was investigated by inhibiting the catalytic activity of enzymes involved in the digestion of disaccharides and polysaccharides to reduce glucose release in the body. Cerium oxide nanoparticles act as a competitive inhibitor in the enzymatic reaction and assume the role of antienzyme. This reduces the enzyme-substrate reaction and prevents enzymatic degradation (61). In the experimental study on enzymes, cerium oxide created strong inhibition on alpha-amylase, alpha-glucosidase and sucrase. These nanoparticles have a high affinity for glucose molecules, similar to metformin. It was discovered that it is useful in regulating glucose metabolism and that the underlying molecular mechanism for the treatment of hyperglycemia is found (62).
This explanation is added to the final part of the discussion in references 61 and 62.
- In our study, hematological parameters were not studied. Almost all studies on cerium oxide nanoparticles emphasized the protective potential of the particles. However, it was reported that chronic exposure, although few, could cause cytotoxicity due to ROS production. Taking advantage of this feature, bactericidal and cancer cell activity was also questioned. However, since our study was applied in a short-term low dose, cytotoxicity was not questioned.
- Cerium oxide has been used in studies for ischemia-reperfusion, sepsis and antibactericidal activity in different models. Since cerium oxide affects damage mechanisms that concern all systems such as oxidative stress and inflammation, no organ-specific activity has been mentioned so far.
Reviewer 2 Report
Comments and Suggestions for Authors
Congratulations for your interesting work. Some suggestions for improvement:
- Please fix the manuscript within the page limits, as some parts of the manuscript are mixed with the numbers on the right side of the manuscript. Please make the size of the letters of the manuscript parts uniform. Please fix the numbers of the citations so that they can be properly presented within the page limits
- I could not find the figures of the manuscript. Are they included in a separate file? As I did not see any figures in the manuscript I have downloaded. Also, please make sure that you should present tables and legends at the end of the manuscript. Did you not have to present them within the text?
- In the abstract you mention that you have 4 groups but you did not mention the results regarding the IR-CO group at all. Please fix that
- At the end of the introducion please mention that the study regarded rats (lines 77-78)
- Regarding the experimental protocol, did you use the methodology or any standard doses for the drugs administered according to other published data of you had your own protocol according to your previous experience? Please make that clearer
- Do you believe that 24 subjects divided in 4 groups are adequate to extract reliable results, especially when the results regard fluctuations in serum particles? I believe that this is the major limitation of your study (very small number of subjects) and should be mentioned in the limitations paragraph.
- In the tables please mention all the abbreviations.
Author Response
- Manuscript corrected within page boundaries. Letters were made uniform in size.
- All figures have been uploaded to the system. Be sure to provide tables and explanations. Due to the writing rules, figures and tables can be at the end of the manuscript.
- The necessary explanation about the CO-IR group has been added to the Abstract section."Renal and liver histopathological findings decreased significantly in the CO-IR group compared to the IR group. A decrease in the TOS level and an increase in the TAS level were found compared to the IR group."
- At the end of the introduction it is stated that the study involved rats.
- In accordance with the literature, similar to previous studies with cerium oxide, a dose of 0.5 mg/kg was used in our study. The myocardial ischemia reperfusion model has also been performed more than once by our team. We share with you studies with similar doses and methodologies.
-The Effects of Cerium Oxide on Sevoflurane Anesthesia and its Relationship to Renal Injury in Rats Seryum Oksidin Sevofluran Anestezisi Üzerine Etkileri ve Böbrek Hasarı ile İlişkisi," Gazi Medical Journal , vol.34, no.3, pp.283-287, 2023
- Manne ND, Arvapalli R, Nepal N, et al. Therapeutic Potential of Cerium Oxide Nanoparticles for the Treatment of Peritonitis Induced by Polymicrobial Insult in Sprague-Dawley Rats. Crit Care Med. 2015;43(11):e477-e489. doi:10.1097/CCM.0000000000001258
- Z. A. ÖZDEMİRKAN Et Al. , "Effect of Cerium Oxide on Kidney and Lung Tissue in Rats with Testicular Torsion/Detorsion," BIOMED RESEARCH INTERNATIONAL , vol.2022, 2022
- Özer A, Şengel N, Küçük A, et al. The Effect of Cerium Oxide (CeO2) on Ischemia-Reperfusion Injury in Skeletal Muscle in Mice with Streptozocin-Induced Diabetes. Medicina (Kaunas). 2024;60(5):752. Published 2024 Apr 30. doi:10.3390/medicina60050752
-Gülcan MB, Demirtaş H, Özer A, et al. Ozone Administration Reduces Myocardial Ischemia Reperfusion Injury in Streptozotocin Induced Diabetes Mellitus Rat Model. Drug Des Devel Ther. 2024;18:4203-4213. Published 2024 Sep 20. doi:10.2147/DDDT.S482309
- ÖZER Et Al., "The Effect of Dexmedetomidine on Ischemia Reperfusion Injury in Myocard of Rat," GAZI MEDICAL JOURNAL, vol.29, no.1, pp.53-56, 2018
- According to the ARRIVE guideline for animal experiments, the use of the smallest possible number of rats in the study is recommended. We conducted the study with the same number of rats in accordance with this guideline and published them in Q1-Q2-Q3 journals. In order for the results to be correct, the collection of blood samples, storage conditions, centrifugation and biochemical analysis were performed repeatedly by the same people in our team. For this reason, we believe that 24 rats will be sufficient.
- All abbreviations are indicated in the tables
Round 2
Reviewer 2 Report
Comments and Suggestions for Authors
Authors made most changes reviewer asked for. I would suggest that you mention the small number of subjects as a limitation of your study in the respective paragraph of the discussion. Otherwise, congratulations for your work.
Author Response
Dear Editor,
We thank the reviewers for their valuable opinions and contributions to our study. We carefully evaluated the valuable reviewer opinions and made the necessary changes to our manuscript according to these suggestions. We indicate the changes made below.
Reviewer (Green)
-Authors made most changes reviewer asked for. I would suggest that you mention the small number of subjects as a limitation of your study in the respective paragraph of the discussion. Otherwise, congratulations for your work.
-Added